# *Eimeria tenella* AMA1 regulates host cell apoptosis through the mitochondrial pathway and the death receptor pathway

Xiaoling Lv,[1] Liwen Wang,[1] Sichen Dong,[1] Yutong Yang,[1] Xueqi Zhang,[1] Tong Xu,[1] Kailing Cui,[1] Zhibin Niu,[1] Wenrui Zhao,[1] Rui Bai,[1] Li Zhang,[1] Mingxue Zheng[1]

**ABSTRACT** *Eimeria tenella* can cause severe damage to the organism by regulating host cell apoptosis during parasitic infection. *E. tenella* apical membrane antigen 1 (*Et*AMA1) is integral to the invasion process and intracellular survival of *E. tenella*. To investigate whether *Et*AMA1 affects host cell apoptosis during parasite infection, we first constructed an *Et*AMA1 expression system. Western blot and indirect immunofluorescence analyses were performed using polyclonal antibodies against the EtAMA1 protein to evaluate its expression and localization. The impact of *Et*AMA1 on host cell apoptosis was examined using Annexin V-fluorescein isothiocyanate/propidium iodide fluorescence staining, quantitative PCR, Western blot, transmission electron microscopy, and other methods. The role of *Et*AMA1 in two apoptotic pathways was further explored by treating host cells with inhibitors targeting either the mitochondrial or death receptor pathways. The findings revealed that *Et*AMA1, a naturally occurring protein with cell proliferation activity in the parasite, was successfully expressed and was shown to inhibit host cell apoptosis via both the death receptor and mitochondrial pathways. These results suggest that *E. tenella* regulates host cell apoptosis through *Et*AMA1.

**IMPORTANCE** *E. tenella* can regulate host cell apoptosis through *Et*AMA1. The inhibitory effect of *Et*AMA1 on apoptosis is achieved through the mitochondrial and death receptor apoptotic pathways. This study can provide new ideas and theoretical support for the research of coccidiostats and vaccines.

**KEYWORDS** *Eimeria tenella*, *Et*AMA1, protein expression, apoptosis

Chicken coccidiosis is an intracellular parasitic protozoan disease with global prevalence (1). The incidence rate in intensive farms typically ranges from 20% to 80%, with mortality rates between 60% and 80% (2). This disease causes an annual global economic loss estimated at 10.4 billion pounds (3). In China alone, over 1 billion yuan is spent each year on drugs to prevent and control coccidiosis. Current prevention and control strategies rely heavily on coccidiostats and vaccines (4). However, drug-based approaches can lead to issues such as drug resistance and immunosuppression, while vaccines, despite their effectiveness, are costly to produce and may negatively affect chick weight gain (5). These measures do not fully address the problem, emphasizing the need for a thorough understanding of the signaling molecules responsible for host cell damage caused by coccidia (6).

*Eimeria tenella* is a highly pathogenic and widely distributed species among known chicken coccidia (7). After infection, parasites typically influence host cell apoptosis to support their growth, development, and reproduction (8). Apoptosis, arising from interactions between parasites and host cells, is a critical pathogenic mechanism of *E. tenella* (9). Interestingly, the parasite's effect on apoptosis varies across different stages of its life cycle. During the early stages of infection, *E. tenella* suppresses host cell apoptosis,

Address correspondence to Mingxue Zheng, zhengmingxue288@163.com.

The authors declare no conflict of interest.

See the funding table on p. 15.

whereas in later stages, apoptosis is promoted (10). Notably, during the schizogenic stage, apoptosis in cecal mucosal epithelial cells significantly increases. This inhibition of host cell apoptosis in earlier stages is thought to facilitate intracellular growth and development (11). Previous studies have shown that *E. tenella* can induce apoptosis via the mitochondrial pathway, as demonstrated by inhibition and protection experiments (8). Activation of Caspase-9 and Caspase-3 plays a central role in mitochondrial apoptosis within host cells (12). Additionally, *E. tenella* regulates the death receptor pathway by regulating the activation of Caspase-8 and Caspase-3 (13). Together, these findings indicate that *E. tenella* influences host cell apoptosis through both mitochondrial and death receptor pathways. However, the specific protein or proteins within the parasite responsible for regulating these pathways remain unidentified.

*Eimeria tenella* is a representative protozoan of the phylum Apicomplexa, which includes species such as *Plasmodium* spp., *Toxoplasma gondii*, and *Neospora caninum* (13). Virulence proteins secreted by members of this phylum, including microneme proteins, rhoptry proteins, and dense granule proteins, play significant roles in regulating host cell apoptosis (14). For example, *Toxoplasma gondii* rhoptry protein 16 directly activates signal transducers and activators of transcription (STAT3/6), thereby inhibiting host cell apoptosis (15, 16). Similarly, *E. tenella* hoptry protein 38 (ROP38) suppresses apoptosis in chicken embryonic cecal epithelial cells, while *Et*ROP30 induces it (17). *Eimeria acervulina* microneme protein 3 inhibits host cell apoptosis by interacting with Casitas B lymphoma protein (18). Moreover, the *E. tenella* MIC4 EGF-like protein reduces cell apoptosis caused by infection through the epidermal growth factor receptor pathway (1).

After adhering to host cells, the top of the parasite forms a tight moving junction (MJ) with the surface of the host cell membrane, which plays a crucial role in the invasion of host cells by the parasite. The *E. tenella* apical membrane antigen 1 (*Et*AMA1), a type of microneme protein, is a key molecule that makes up MJ. The parasite is propelled into the host cell by actin and myosin motors after MJ formation and completed its invasion process. Meanwhile, *Et*AMA1 is also a key virulence factor of the parasite (19). However, whether *Et*AMA1 regulates host cell apoptosis and the underlying mechanisms remain unclear.

In this study, various techniques, including 3-(4,5-dimethylthiazol-2-yl)-2,5-diphenyl-tetrazolium bromide (MTT) assay, quantitative PCR (qPCR), Western blot, fluorescence probes, indirect immunofluorescence, spectrophotometry, and ELISA, were employed to investigate the role and mechanism of *Et*AMA1 in regulating host cell apoptosis. These findings aim to improve the understanding of the function and mechanism of the virulence protein AMA1 in *E. tenella* infection while also offering new insights for developing innovative coccidiostats and vaccines targeting *E. tenella* virulence proteins.

## MATERIALS AND METHODS

### Animals and parasite

Fifteen-day-old specific-pathogen-free (SPF) chicken embryos were obtained from Beijing Vital River Laboratory Animal Technology Co., Ltd. (Beijing, China). The strain used in this study was the Shanxi virulent strain of *Eimeria tenella* (EtSX01), provided by the Veterinary Pathology Laboratory, College of Veterinary Medicine, Shanxi Agricultural University.

### Main reagents

MTT, 4% paraformaldehyde, Triton X-100, isopropyl β-D-1-thiogalactopyranoside (IPTG), Luria-Bertani broth and agar powder, and dimethyl sulfoxide (DMSO) were purchased from Solarbio Life Sciences (Beijing, China); pET-28a(+) DNA, BL21(DE3) competent cells, ampicillin sodium, and the SanPrep Column Plasmid Mini-Preps Kit were from Sangon Biotech (Shanghai, China); the X33 yeast strain and the pPICZαA *Pichia* expression vector

were from MiaoLingBio (Wuhan, China); Zeocin was from Invitrogen (California, USA); The BMGY/BMMY medium base was from Topbio (Shandong, China); the DL5,000 DNA marker, PrimeScript RT reagent kit, TB Green Premix Ex Taq II, and RNAiso Plus were from Takara (Tokyo, Japan); the 6×His and His-tag polyclonal antibodies, as well as Bax polyclonal antibody, were from Proteintech (Chicago, IL, USA); the BD Pharmingen FITC Annexin V Apoptosis Detection Kit I was from BD Biosciences (New Jersey, USA); Z-IETD-FMK and Z-LEHD-FMK were from Selleck (Houston, TX, USA); BeyoBlue Coomassie Blue Super Fast Staining Solution and caspase activity assay kits (Caspase 3, Caspase 8, and Caspase 9) were from Beyotime (Shanghai, China); Fas Rb pAb and TNFR1 Rb pAb were from Wanleibio (Shenyang, China); the Bcl-2 rabbit antibody was from BIOSS (Beijing, China); the goat antirabbit and mouse IgG-HRPs were from Abmart (Shanghai, China); and the hypersensitive enhanced chemiluminescence kit was from Boster (Wuhan, China).

## Cultivation of primary cecal epithelial cells from chicken embryos

Chicken cecal epithelial cells were isolated and cultured following the method described by Yang et al. (20). Briefly, the eggshells of 15-day-old SPF chicken embryos were carefully opened using surgical scissors. The embryos were removed, and the ceca were isolated with tissue scissors. After rinsing with phosphate-buffered saline (PBS) buffer, the ceca were minced into the smallest possible pieces. The tissue fragments were then mixed with thermophilic protease (50 mg/L) and digested at $41°C$ for 2 h. Digestion was halted by adding PBS, followed by centrifugation at 1,200 rpm for 5 min, and the supernatant was discarded. The resulting cell pellet was suspended in low-glucose Dulbecco's modified Eagle's medium (DMEM) cell culture medium containing 10% fetal bovine serum (FBS) and seeded into a cell culture flask for 70 min. Afterward, the suspension was collected, centrifuged again at 1,200 rpm for 5 min, and resuspended in DMEM/F12 medium containing 2.5% FBS. The cells were then inoculated into a cell culture plate. When the cell adhesion rate reached approximately 85%, the cells were deemed ready for subsequent experiments.

## Preparation of *E. tenella* sporozoites

The preparation of *E. tenella* sporozoites followed the protocol described by Yang et al. (20). Sporulated oocysts of *E. tenella* in potassium dichromate solution were centrifuged at 2,000 rpm for 5 min, and the supernatant was discarded. The oocyst pellet was thoroughly washed with PBS and resuspended in 2 mL PBS. The suspension was then ground using a grinder until the excystation rate reached 80%. The mixture was centrifuged at 1,800 rpm for 5 min, and the supernatant was discarded. The resulting pellet was treated with sporozoite digestion solution and incubated at $41°C$ with shaking at 150 rpm. Digestion was stopped when 80% of the spores were released. The mixture was then centrifuged at 3,000 rpm for 10 min, and the final precipitate was resuspended in DMEM nutrient solution.

## Construction of PPICZαA-*Et*AMA1 yeast expression system and pET-*Et*AMA1 prokaryotic expression system

The *EtAMA1* gene sequence (GenBank: JN032081) was optimized using Jcat software based on the codon preferences of the *Pichia pastoris* genome. EcoR I and Sal I restriction enzyme cleavage sites were added at both ends of the gene for synthesis and identification. Recombinant plasmids were digested with Sac I enzyme, and the products were recovered. The linearized plasmids were then transformed into X-33 *Pichia pastoris* competent cells using electroporation with the following parameters: voltage, 1,500 V; resistance, 200 Ω; capacitance, 25 μF; and pulse time, 4–5 ms. The transformed cells were cultivated for 3–6 days until white monoclonal colonies appeared. A single colony was selected and cultured in 10 mL BMGY medium at 28 °C and 200 rpm for 16–18 h until it reached the logarithmic growth stage. The bacterial cells were collected by

centrifugation at room temperature at 1,500 rpm for 5 min, and the cells were resuspended in 12.5 mL of BMMY medium until an $OD_{600} = 1$ was obtained. The cells were cultivated at 26 °C, 200 rpm, and methanol containing 100 mM protease inhibitor was added every 12 h to achieve a final concentration of 1%. Culture medium (0.5 mL) was taken at 24, 48, 72, and 96 h, respectively, and protein expression levels were detected to determine protein expression conditions. According to the protein expression conditions, the recombinant protein EtAMA1 was expressed, purified, and identified.

To construct the prokaryotic expression system, the *EtAMA1* gene was amplified from *E. tenella* genomic DNA by PCR and subcloned into the pET-28a(+) DNA expression vector. A single colony was selected and cultured at 37 °C and 200 rpm until the $OD_{600}$ was around 0.6–0.8. IPTG was added at a final concentration of 1 mmol/L, and expression was induced at 30°C and 200 rpm. One milliliter of bacterial solution was taken at 2, 4, 6, and 8 h after induction, and protein expression levels were detected to determine protein expression conditions. According to the protein expression conditions, the recombinant protein was expressed, purified, and identified.

## Preparation of *E. tenella*-positive and rEtAMA1-positive serum

Fourteen-day-old SPF chickens were inoculated with $1 \times 10^4$ *E. tenella* sporulated oocysts. Subsequently, 5,000 sporulated oocysts were administered every 3 days, for a total of four inoculations. Blood was collected via cardiac puncture, left to stand at room temperature for serum separation, and the *E. tenella*-positive serum was stored at −80°C.

Purified *Et*AMA1 recombinant protein was emulsified with an equal volume of complete Freund's adjuvant. The first immunization used complete Freund's adjuvant emulsified with recombinant protein, while subsequent inoculations employed incomplete Freund's adjuvant emulsified with recombinant protein. Three 2-month-old male New Zealand white rabbits were immunized with a 200 µg dose of recombinant protein every 2 weeks. Blood was collected from the ear vein before each immunization to monitor antibody titer. After the third immunization, blood was collected via cardiac puncture, and the serum was separated. The serum was inactivated by incubation in a 56°C water bath for 30 min and stored at −80°C.

## Immunofluorescence staining and confocal microscopy

After inoculation with *E. tenella*, the chickens were sacrificed, and cecal tissues were prepared as paraffin sections. The sections were dewaxed, hydrated, and subjected to antigen retrieval. They were then treated with 0.1% Triton X-100 at room temperature for 15 min and washed with PBS. Blocking was performed using 5% bovine serum albumin (BSA) at 37°C for 1 h, followed by washing with PBS. The sections were incubated overnight at 4°C with *Et*AMA1 polyclonal antibody (1:500) and washed four times with PBS. Fluorescein isothiocyanate (FITC)-conjugated goat antirabbit IgG (green) was applied at 37°C for 1 h, followed by washing with PBS. Nuclei were stained with Hoechst, washed, and the slides were sealed with an antifade mounting medium. Images were captured using a confocal microscope.

For sporozoite staining, *E. tenella* spores were placed on glass slides, allowed to dry thoroughly, and fixed with 4% paraformaldehyde at room temperature for 15 min, followed by washing with PBS. The slides were then treated with 0.1% Triton X-100 at room temperature for 15 min and washed with PBS. Blocking was carried out using 5% BSA at 37°C for 1 h, followed by washing with PBS. Staining and observation steps were performed as described above.

## Cytotoxicity assay

Chicken embryo cecal epithelial cells were seeded into 96-well plates. When the cell adhesion rate reached approximately 85%, the cells were treated with *Et*AMA1 recombinant protein or various inhibitors (Z-IETD-FMK and Z-LEHD-FMK) at specified concentrations for 24 h. After washing with PBS, 15 µL of MTT was added to each well, and the

plates were incubated at 41°C for 4 h. Subsequently, 150 μL of DMSO was added to each well. The absorbance was measured using a microplate reader.

## Apoptosis detection

At the designated time of infection, the nutrient solution was discarded, and the cells were washed with PBS. After washing, the cells were digested with trypsin-EDTA solution, and digestion was terminated by adding DMEM containing 10% FBS. The cells were then collected by centrifugation. The pellet was resuspended in 500 μL of binding buffer after washing. The samples were incubated with 5 μL of Annexin V-FITC and 1 μL of propidium iodide at room temperature for 30 min. After incubation, the samples were centrifuged at 1,000 rpm for 10 min, and the cells were resuspended in 50 μL of 1× binding buffer. The samples were sealed with an antifluorescence sealing agent, observed, and photographed under a fluorescence microscope.

### Judgment criteria

Nuclei stained blue with Hoechst staining indicate intact cells. Cells not undergoing apoptosis exhibit low-intensity green and red staining. Cells in ṁ show strong green staining and weak red staining. Cells in late apoptosis or necrosis display strong staining for both green and red. Each time point included five replicates, and at least 200 cells per sample were counted to calculate the apoptosis rate:

Early apoptosis rate $=$ (early apoptotic cells / total observed cells) $\times 100\%$
Late apoptosis and necrosis rate $=$ (late apoptotic and necrotic cells / total observed cells) $\times 100\%$ .

## Caspase-3, Caspase-8, and Caspase-9 activity detection

At the designated time of infection, the nutrient solution was discarded, and the cells were washed with PBS. After washing, the cells were digested with trypsin-EDTA solution, and digestion was terminated using DMEM containing 10% FBS. The cells were collected by centrifugation at 1,500 rpm at 4°C for 5 min. For every 2 million cells, 100 μL of lysis solution was added, and the samples were incubated on ice for 15 min. The lysates were centrifuged at 12,000 × $g$ at 4°C for 15 min, and the supernatant was collected. The activities of Caspase-3, Caspase-8, and Caspase-9 were measured according to the protocols provided with the detection kits.

## qPCR analysis

Total RNA was extracted from chicken embryo cecal epithelial cells infected with *E. tenella* using TRIzol reagent. cDNA was synthesized using PrimeScript reverse transcriptase. The mRNA levels of Caspase-3, Caspase-8, Caspase-9, Bcl-2, Bax, TNFR1, Fas, and TRAIL were quantified by fluorescence-based qPCR. The primer sequences used in the analysis are listed in Table 1.

## Western blot

At the designated time of infection, the nutrient solution was discarded, and the cells were washed with PBS. The cells were lysed on ice for 30 min using radioimmunoprecipitation assay buffer. The lysates were centrifuged at 12,000 × $g$ at 4°C for 15 min. Protein samples (50 μg per lane) were separated using 12% polyacrylamide gels and subsequently transferred to 0.22 μm polyvinylidene fluoride membranes. The membranes were blocked with 5% nonfat dry milk at 37°C for 1 h and then incubated overnight at 4°C with specific primary antibodies. After washing with PBST, the membranes were incubated at 37°C for 1 h with species-specific HRP-conjugated secondary antibodies. The membranes were washed again with PBST before exposure, and images were captured. β-Actin was used as an internal control.

**TABLE 1** Primer sequences for qPCR

| Gene | Primer sequence | Length（bp） | GenBank ID |
|---|---|---|---|
| β-Actin | 5′-CACCACAGCCGAGAGAGAAAT-3′ | 135 | L08165.1 |
| | 5′-TGACCATCAGGGAGTTCATAGC-3′ | | |
| Caspase-3 | 5′-ATTGAAGCAGACAGTGGACCAGATG-3′ | 111 | NM_204725.2 |
| | 5′-TGCGTTCCTCCAGGAGTAGTAGC-3′ | | |
| Caspase-8 | 5′-CATCTGTGGCACCCGATTCTCTG-3′ | 148 | NM_204592.4 |
| | 5′-CTTCTGAGTTCTGGCACTGCTTCC-3′ | | |
| Caspase-9 | 5′-CGCTTGTCCATCCCAGTCCAAC-3′ | 97 | XM_424580.7 |
| | 5′-CCAGTCTGTGGTCACTCTTGTCAAC-3′ | | |
| Bcl-2 | 5′-ACCTGGATGACCGAGTACCTGAAC-3′ | 92 | Z11961.1 |
| | 5′-CTGTTGCCGTACAATTCCACAAAGG-3′ | | |
| Bax | 5′-ATGGAGGTGATGGAGGTGAC-3′ | 143 | FJ977571.1 |
| | 5′-TCCTGGTCGAGTTTGTCACC-3′ | | |
| TNFR1 | 5′-ATACTGTGTGTGGCTGTCGG-3′ | 205 | XM 015294558.1 |
| | 5′-AAGCACTCTTCTCCAACGCA-3′ | | |
| Fas | 5′-ACCGTCTATCGGAACCTACCA-3′ | 126 | AF 296875.1 |
| | 5′-AGGTTTCGTAGGCTCCTCCC-3′ | | |
| TRAIL | 5′-GCGTCCCCGCACATAGATTA-3′ | 206 | NM 204379.2 |
| | 5′-AACAAACCCGAGTCCTCGTC-3′ | | |

## Images and statistical analyses

All experiments in this study were repeated three or more times. Data analysis was performed using analysis of variance in SPSS version 26.0 statistical software (Chicago, USA). Results are presented as mean ± SD. Histograms were generated using Graph-Pad Prism version 5.0 software (San Diego, CA, USA). A $P$ value of less than 0.05 was considered statistically significant.

## RESULTS

### The expression of *Et*AMA1 recombinant protein

The expression of the *Et*AMA1 recombinant protein in yeast was induced using methanol. Specific protein bands were observed at ~58 kDa during induction at 24, 48, 72, and 96 h (Fig. S1). Western blot analysis confirmed the presence of bands at ~58 kDa post-induction, with the highest eukaryotic recombinant protein expression observed at 48 h (Fig. S1B). Following nickel column purification, the recombinant eukaryotic protein was determined to be pure and suitable for subsequent studies (Fig. S1C and D). Primary chicken embryonic cecal epithelial cell activity increased with higher *Et*AMA1 concentrations, peaking at 0.1 µg/mL. Cell proliferation activity was significantly greater than that in the 4 and 48 h and control groups when treated with 0.1 µg/mL for 24 h (Fig. S1E). These results indicate that an *Et*AMA1 protein expression system with cell proliferation activity has been successfully established.

The *Et*AMA1 protein expression in the prokaryotic recombinant vector was induced using IPTG. A target protein band was observed at ~58 kDa in the supernatant, but no such band was detected in the sediment (Fig. S2A). These findings suggest that the prokaryotic recombinant *Et*AMA1 protein was expressed in a soluble form. Coomassie brilliant blue staining and Western blot analysis demonstrated that the highest expression of prokaryotic recombinant protein occurred at 6 h post-induction (Fig. S2B and C). After nickel column purification, the prokaryotic recombinant protein was confirmed to be pure and suitable for further studies (Fig. S2D and E).

### *Et*AMA1 reactivity and localization identification

Rabbit serum titers exceeded $1:2.56 \times 10^4$ after three rounds of immunization with the *Et*AMA1 eukaryotic recombinant protein (Table 2). Western blot results showed that

**TABLE 2** Titer determination of *Et*AMA1 polyclonal antibody[a]

| | | | | Antibody dilution ratio | | | | | Negative serum |
|---|---|---|---|---|---|---|---|---|---|
| 1:100 | 1:200 | 1:400 | 1:800 | 1:1,600 | 1:3,200 | 1:6,400 | 1:12,800 | 1:25,600 | |
| * | * | * | 3.371 | 2.772 | 2.046 | 1.526 | 1.054 | 0.720 | 0.360 |

[a] "*" indicates an $OD_{450}$ value exceeding 3.5.

chicken anti-*E. tenella*-positive serum reacted with the *Et*AMA1 eukaryotic recombinant protein, whereas negative serum did not (Fig. 1A). This indicates that *Et*AMA1 is a natural protein of *E. tenella*. Indirect immunofluorescence using rabbit polyclonal antibodies as primary antibodies demonstrated that *Et*AMA1 was localized on the surface and inside schizonts and sporozoites (Fig. 1B and C).

## *Et*AMA1 eukaryotic recombinant protein inhibits host cell apoptosis

The pattern of host cell apoptosis in the *E. tenella* group aligned with previously reported findings. Specifically, *E. tenella* inhibited apoptosis in the early stages (at 4 h post-infection) but promoted apoptosis at later stages (from 24 to 96 h post-infection). At 4 h post-infection, the apoptosis rate in the *Et*AMA1 group was comparable to that of the control group. However, the apoptosis rate in the *E. tenella* + *Et*AMA1 group was significantly lower than those in other groups. From 24 to 96 h post-infection with *E. tenella*, the apoptosis rate in the *Et*AMA1 group was significantly lower than those in other groups. During the same period, the apoptosis rate in the *E. tenella* + *Et*AMA1 group was higher than that in the control group but lower than that in the *E. tenella*

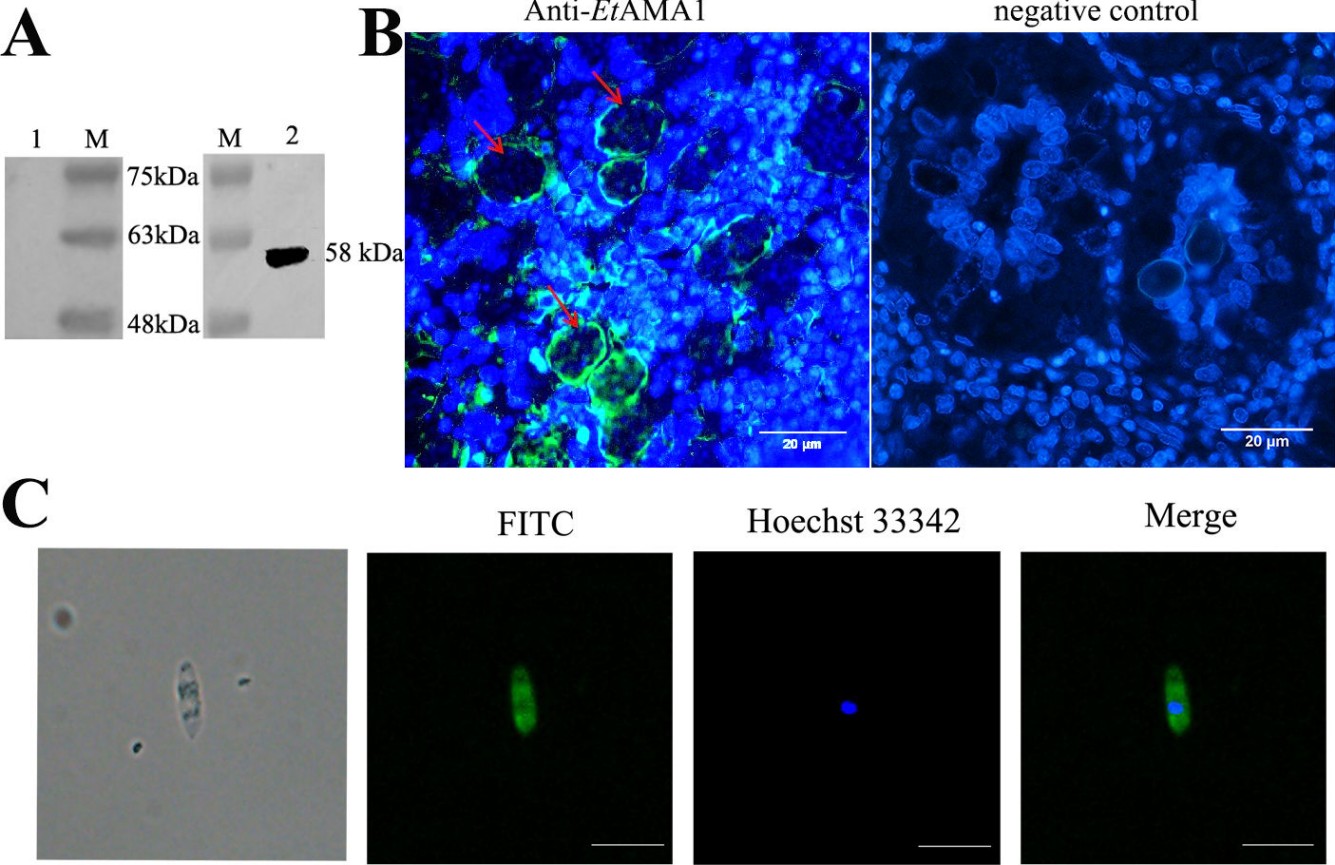

**FIG 1** *Et*AMA1 reactivity and localization identification. (A) Western blotting results of *Et*AMA1 protein. M, marker; 1, negative control; 2, *Et*AMA1 recombinant protein. (B) *Et*AMA1 protein expression in *E. tenella* schizozoite stage. Note: Arrow indicates schizont (×400). (C) Results of *Et*AMA1 protein expression at the sporozoite stage of *E. tenella*. Scale bar = 20 µm.

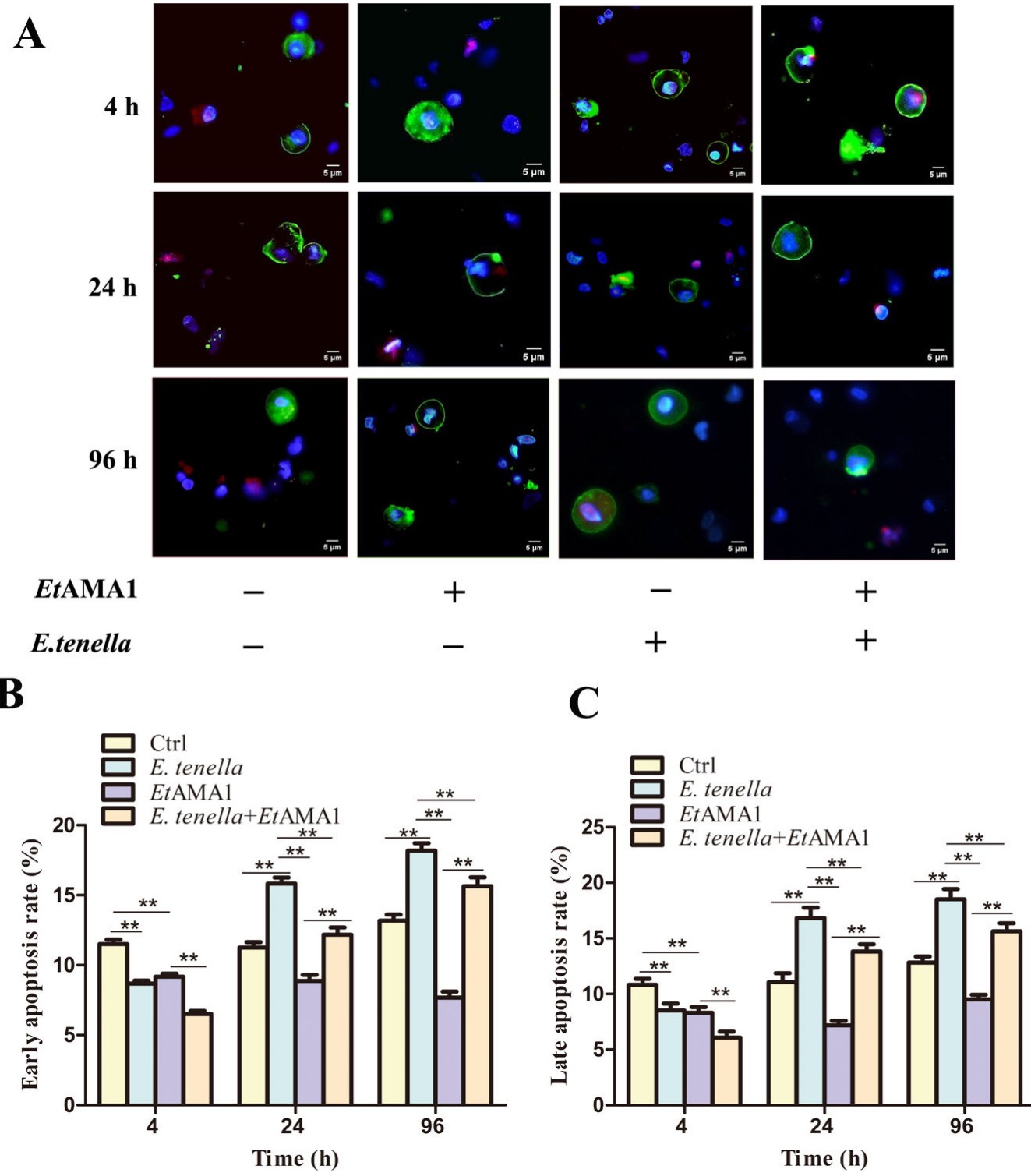

**FIG 2** The effect of *Et*AMA1 on host cell apoptosis. (A) Effect of *Et*AMA1 protein on host cell apoptosis after *E. tenella* inoculation for 4–96 h. Early apoptotic cells were highly stained with Annexin V-FITC (green) and weakly stained with propidium iodide (PI) (red). Advanced apoptotic and necrotic cells are highly stained with Annexin V-FITC and PI. (B and C) Quantitative detection of apoptosis at different time points after infection. Each value represents the average value from 30 to 40 cells from at least five regions of three representative experiments. **$P < 0.01$ vs. other group.

group (Fig. 2). These results suggest that host cell apoptosis decreases during the early stage of *E. tenella* infection but increases at later stages. Moreover, *Et*AMA1 significantly inhibits host cell apoptosis.

## *Et*AMA1 regulates Caspase-8, Caspase-9, and Caspase-3 activities

At 4 h post-infection, the mRNA expression and activity levels of Caspase-8, Caspase-9, and Caspase-3 in the *E. tenella* group were noticeably lower than those in the control group. Between 24 and 96 h post-infection, the mRNA expression and activity levels of these three genes in the *E. tenella* group were significantly higher than in other groups. Following *E. tenella* infection, the mRNA expression and activity levels of Caspase-8, Caspase-9, and Caspase-3 in the *Et*AMA1 group were substantially lower than those in the control group. Additionally, the mRNA expression and activity levels of these genes in the *E. tenella* + *Et*AMA1 group were significantly lower than those in the *E. tenella* group (Fig. 3). These findings indicate that *Et*AMA1 can regulate the activity of key executors and effectors in both mitochondrial and death receptor apoptotic pathways.

## Caspase-8 and Caspase-9 inhibitors reduce the inhibitory effect of *Et*AMA1 on host cell apoptosis

Cell proliferation activity was not significantly affected when the Caspase-8 inhibitor (Z-IETD-FMK and IETD) concentration was ≤30 μM compared with the control group. However, when the IETD concentration reached 40 μM, cell proliferation activity in the treated group was significantly lower than that in the control group. A similar pattern was observed with Z-LEHD-FMK (LEHD), where cell proliferation activity followed a comparable trend to that of IETD (Fig. 4A). Based on these findings, the optimal inhibitor concentration for further experiments was determined to be 30 μM.

At 4 h post-infection, the apoptosis rate in the IETD group was the highest among the four groups. The *E. tenella* + IETD + *Et*AMA1 group exhibited a significantly lower apoptosis rate compared to the *E. tenella* + IETD group and the *Et*AMA1 + IETD group. No significant difference in apoptosis rates was observed between the *E. tenella* + IETD group and the *Et*AMA1 + IETD group. From 24 to 96 h, the apoptosis rate in the IETD group was significantly lower than that in the *E. tenella* + IETD group but higher than that in the *Et*AMA1 + IETD group. The *E. tenella* + IETD group showed a significantly higher apoptosis rate than the *Et*AMA1 + IETD group and the *E. tenella* + IETD + *Et*AMA1 group. Additionally, the apoptosis rate in the *E. tenella* + IETD + *Et*AMA1 group was significantly higher than that in the *Et*AMA1 + IETD group (Fig. 4B through D). The effect of LEHD on host cell apoptosis rates mirrored the observations for IETD, with a consistent pattern of changes (Fig. 5). These results demonstrate that *Et*AMA1 inhibits host cell apoptosis by regulating the executors of the mitochondrial and death receptor pathways.

## The effect of *Et*AMA1 on critical factors in the apoptotic pathway of death receptors and mitochondrial pathway

At 4 h post-infection with *E. tenella*, Fas and TNFR1 expression in host cells of the *E. tenella* group was reduced, while TRAIL expression showed no significant change. From 24 to 96 h post-infection, TNFR1 and TRAIL expression levels in host cells were elevated in the *E. tenella* group. In contrast, the *E. tenella* + *Et*AMA1 group showed decreased Fas and TNFR1 expression throughout the infection period (Fig. 6A through C and E). At 4 h post-infection, host cells in the *E. tenella* group exhibited an increase in the Bcl-2:Bax ratio. However, from 24 to 96 h post-infection, the Bcl-2:Bax ratio in the host cells of the *E. tenella* group decreased. In the *E. tenella* + *Et*AMA1 group, the Bcl-2:Bax ratio was consistently higher across all time points (Fig. 6D and E). These findings suggest that *Et*AMA1 regulates host cell apoptosis through both the death receptor and mitochondrial pathways.

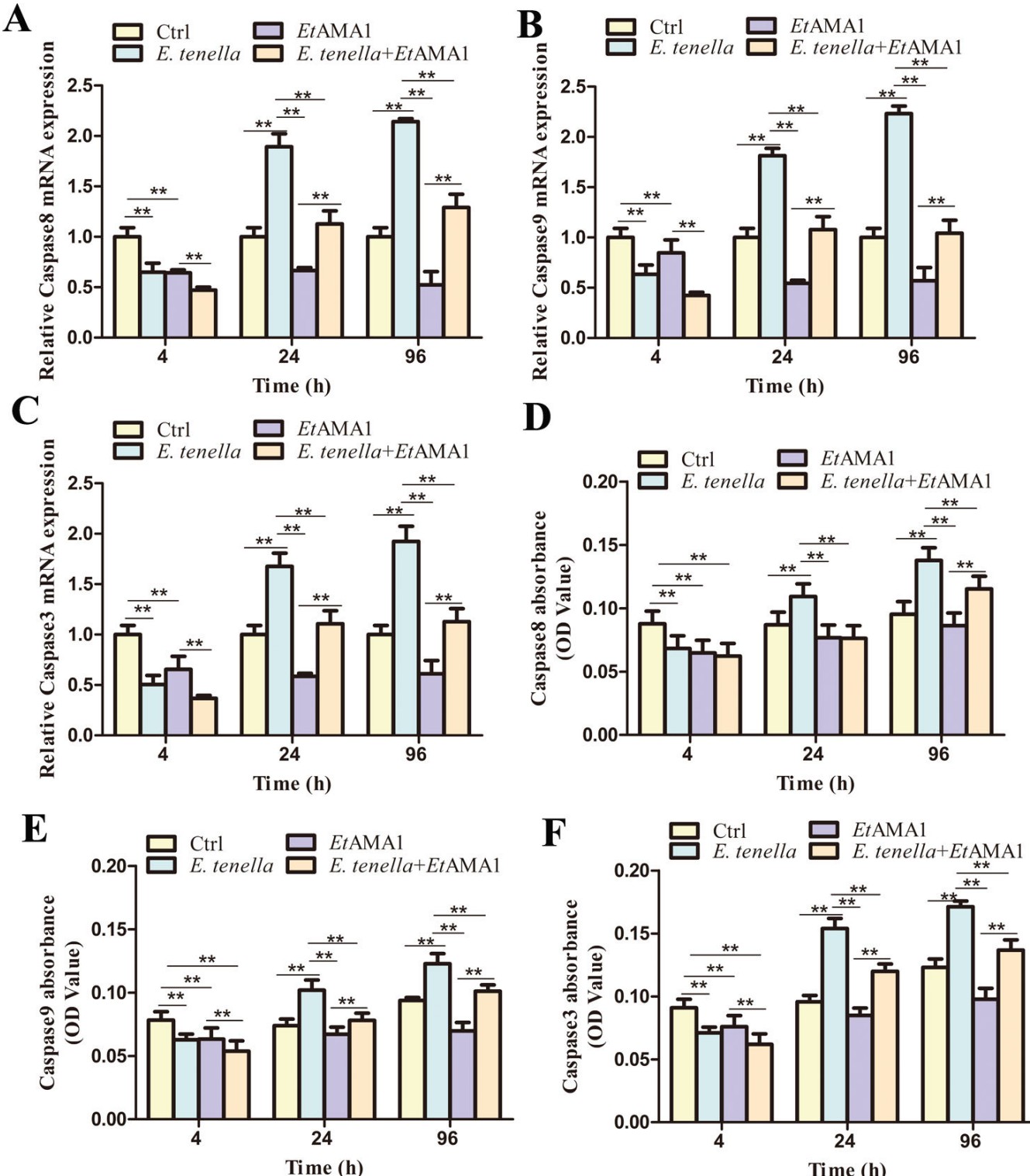

**FIG 3** The effect of *Et*AMA1 on Caspase-3, Caspase-8, and Caspase-9. (A–C) Effect of *Et*AMA1 protein on Caspase-3, Caspase-8, and Caspase-9 mRNA expression. (D–F) Effect of *Et*AMA1 protein on Caspase-3, Caspase-8, and Caspase-9 activity. The experiment was repeated three times. Data analysis was performed using analysis of variance, and results are presented as mean ± SD. **$P < 0.01$ vs. other group.

Overall, these results demonstrate that *Et*AMA1 inhibits apoptosis in *E. tenella* host cells by downregulating the expression of Fas and TNFR1 in the death receptor pathway,

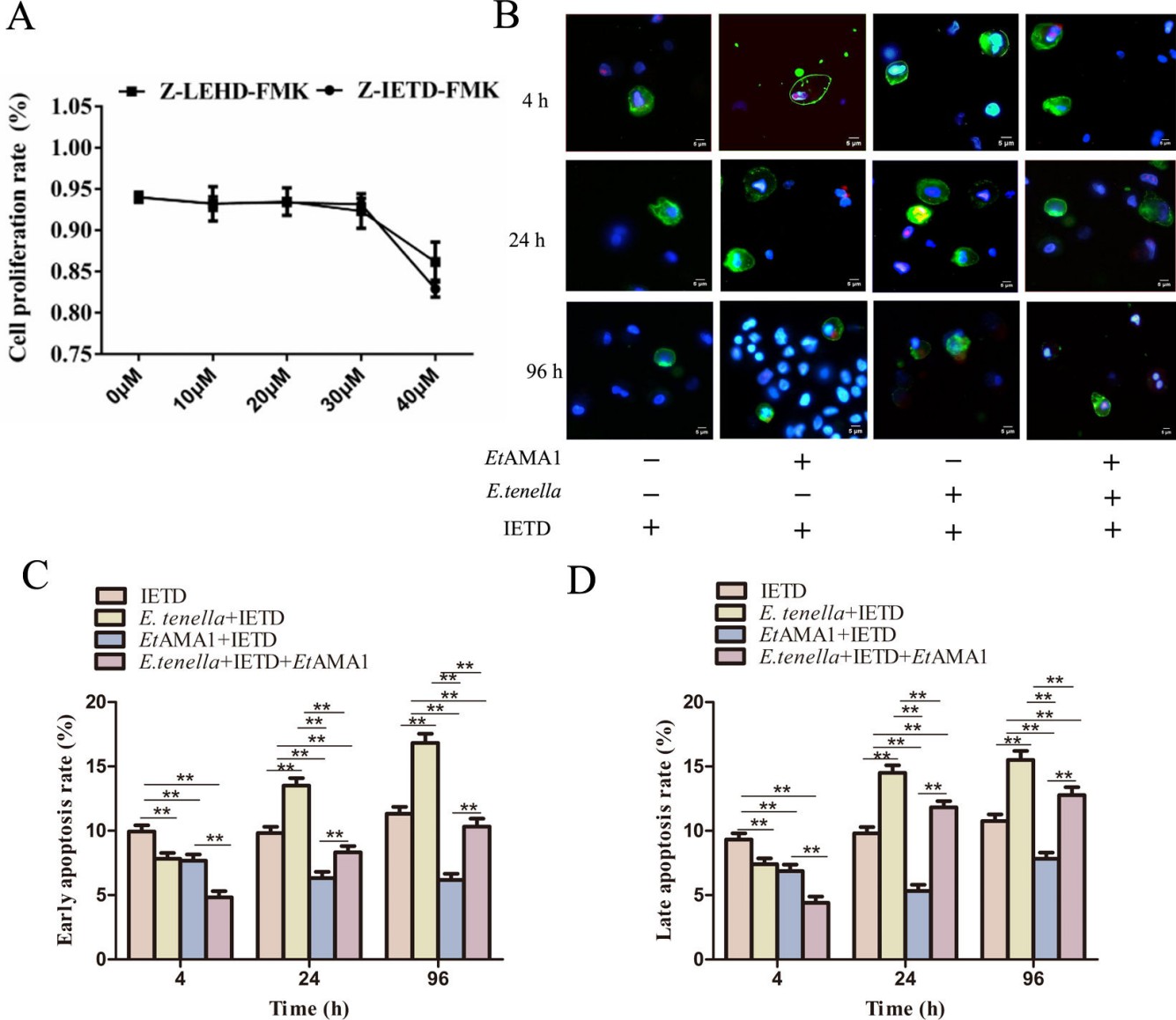

**FIG 4** Caspase-8 inhibitors reduce the inhibitory effect of *Et*AMA1 on host cell apoptosis. (A) Determination of working concentrations of chemical synthesis inhibitors Z-LEHD-FMK and Z-IETD-FMK. (B) Effect of Z-IETD-FMK on host cell apoptosis after *E. tenella* inoculation for 4–96 h. Early apoptotic cells were highly stained with Annexin V-FITC (green) and weakly stained with PI (red). Advanced apoptotic and necrotic cells are highly stained with Annexin V-FITC and PI. (C and D) Quantitative detection of apoptosis at different time points after infection. Each value represents the average value from 30 to 40 cells from at least five regions of three representative experiments. **$P < 0.01$ vs. other group.

reducing the mRNA expression and activity of Caspase-8 and Caspase-3, increasing the Bcl-2:Bax ratio in the mitochondrial pathway, and decreasing the mRNA expression and activity of Caspase-9 and Caspase-3.

## DISCUSSION

*Eimeria tenella* is the most pathogenic species among chicken coccidia, causing significant damage to the mucosa of the cecum and adjacent intestinal segments in chickens (21). Infected chickens typically exhibit mucosal swelling, necrosis, disintegration, and shedding of epithelial cells and microvilli (22). Currently, the primary methods for preventing and controlling chicken coccidiosis are drugs and vaccines. However, issues such as resistance and residues associated with anticoccidial drugs are unavoidable (23). Although live coccidial vaccines provide effective immune protection,

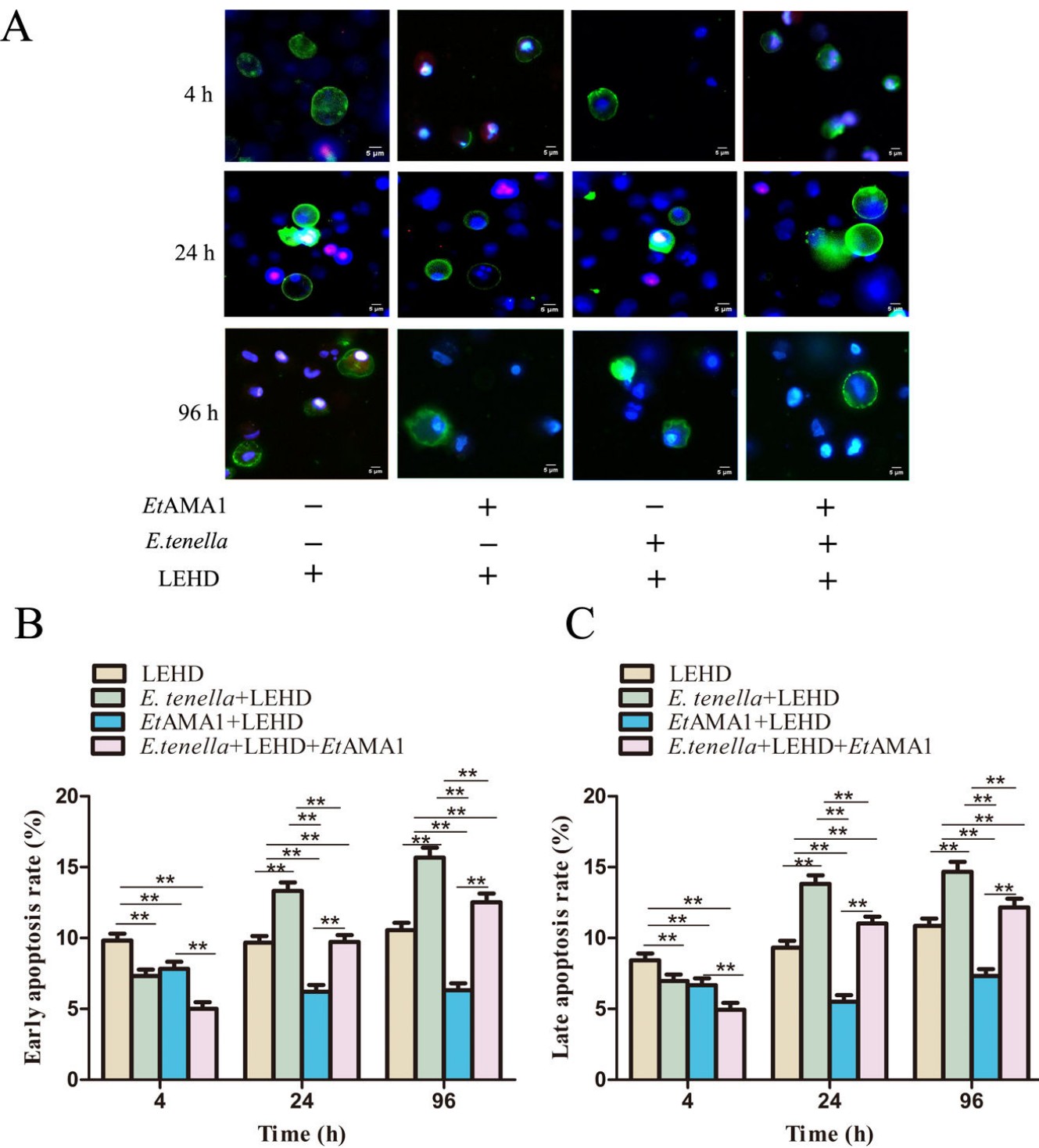

**FIG 5** Caspase-9 inhibitors reduce the inhibitory effect of *Et*AMA1 on host cell apoptosis. (A) Effect of Z-LEHD-FMK on host cell apoptosis after *E. tenella* inoculation for 4–96 h. Early apoptotic cells were highly stained with Annexin V-FITC (green) and weakly stained with PI (red). Advanced apoptotic and necrotic cells are highly stained with Annexin V-FITC and PI. (B and C) Quantitative detection of apoptosis at different time points after infection. Each value represents the average value from 30 to 40 cells from at least five regions of three representative experiments. **$P < 0.01$ vs. other group.

they are associated with problems such as intestinal damage and virulence reversion (19). The antigenic complexity of coccidia, coupled with variations in morphology, structure, and composition across developmental stages, limits the protective efficacy

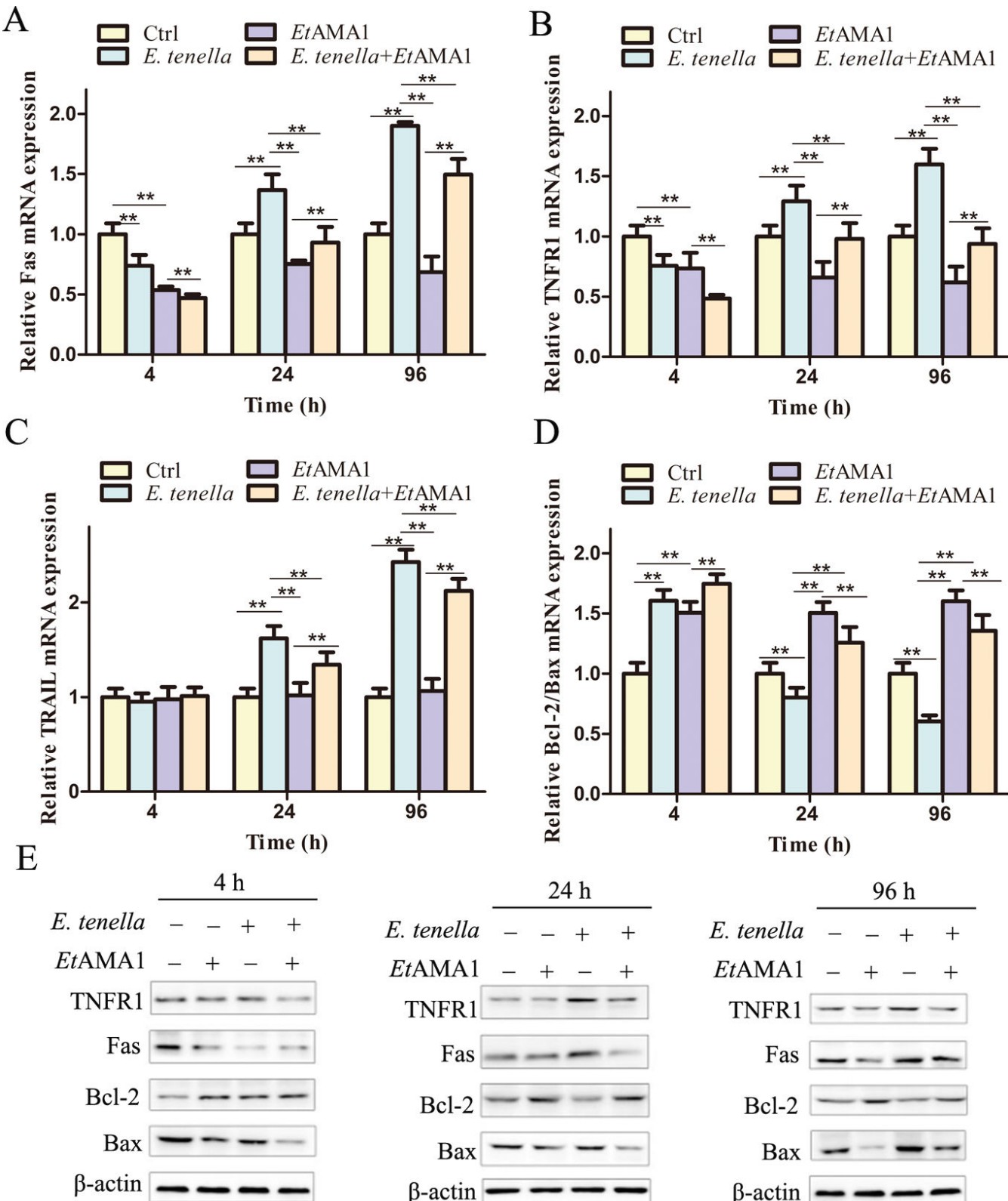

**FIG 6** The effect of *Et*AMA1 on key factors in the apoptotic pathway of death receptors and mitochondrial pathway. (A–D) The mRNA expression of *E. tenella* host cell death receptor and mitochondrial pathway-related genes at 4–96 h after inoculation. **$P < 0.01$ vs. other group. (E) The protein expression of *E. tenella* host cell death receptor and mitochondrial pathway-related genes at 4–96 h after inoculation.

of genetically engineered vaccines (24). Therefore, studying the mechanisms of intestinal injury caused by *E. tenella* is essential for reducing the pathogenicity of attenuated live vaccines and developing novel gene-deficient vaccines.

Apoptosis plays a critical role in the interaction between hosts and parasites (25). Early apoptosis of host cells can help resist intracellular parasitic infections, but it may also facilitate parasite invasion. Conversely, late apoptosis is beneficial for clearing damaged cells, reducing excessive inflammatory responses, and limiting effective protective immune responses (25, 26). Inhibiting apoptosis in infected cells can prevent further spread of intracellular parasites (27). This study confirmed that *E. tenella* infection inhibits apoptosis in primary chicken embryo cecal epithelial cells during the early stages of infection while promoting apoptosis in the later stages.

*E. tenella* regulates host cell apoptosis through both death receptor and mitochondrial pathways to support its development and proliferation. This regulation is a primary cause of the severe cecal damage observed in infected chickens (12, 28). However, the specific proteins involved in this regulation remain unclear. Parasite-secreted virulence proteins are crucial mediators of host damage (29). Micronemes, small tubular organelles unique to apicomplexan protozoa, secrete virulence proteins that play a vital role in host cell invasion (30, 31). *Et*AMA1, a microneme-secreted virulence protein, is expressed at all stages of the *E. tenella* life cycle and is essential for parasite invasion and intracellular survival (32).

In this study, we successfully expressed biologically active *Et*AMA1 eukaryotic recombinant protein using a yeast expression system and confirmed that *Et*AMA1 naturally exists in *E. tenella*. The *Et*AMA1 protein was found to be distributed on the surface and inside schizonts and sporozoites, consistent with the findings of Jiang et al. (33). Both *Et*AMA1 and the inhibitor Z-IETD-FMK or Z-LEHD-FMK were shown to reduce host cell apoptosis rates. Notably, the apoptosis rate was further reduced when *Et*AMA1 was combined with mitochondrial or death receptor pathway inhibitors. These results suggest that *Et*AMA1 or the blockade of the death receptor or mitochondrial pathways can reduce host cell apoptosis induced by *E. tenella* infection. The findings of Li et al. and Xu et al. (8, 9), which independently demonstrated that inhibitors of the mitochondrial or death receptor pathways can lower host cell apoptosis rates, align with the results of this study. Our data indicate that *Et*AMA1 inhibits the mRNA expression of TNFR1, Fas, Caspase-3, Caspase-8, and Caspase-9 in host cells. Additionally, Caspase-3, Caspase-8, and Caspase-9 activities, as well as the protein expression levels of Fas and TNFR1, were reduced, while the Bcl-2:Bax ratio was upregulated. These findings confirm that *Et*AMA1 can inhibit host cell apoptosis by regulating both the death receptor and mitochondrial pathways.

## ACKNOWLEDGMENTS

Our thanks are extended to all the people who made this work possible.

This study was funded by the Shanxi Provincial Key Research and Development Program (grant no. 2022ZDYF126), the Scientific Research Project of Shanxi Province Outstanding Doctoral Work Award Fund (grant no. SXYBKY2019023), and Innovation Projects of College of Veterinary Medicine, Shanxi Agricultural University (grant no. J202111305).

X.L. carried out most of the experiments, wrote the manuscript, and should be considered the primary author. M.Z. critically revised the manuscript and the experiment design. S.D., L.W., X.Z., Y.Y., T.X., K.C., Z.N., W.Z., R.B., and L.Z. helped with the experiment. All the authors read and approved the final version of the manuscript.

## AUTHOR AFFILIATION

[1]College of Veterinary Medicine, Shanxi Agricultural University, Taigu, Jinzhong, China

## AUTHOR ORCIDs

Xiaoling Lv  http://orcid.org/0000-0003-1676-9917

Mingxue Zheng  http://orcid.org/0000-0003-0142-4377

## FUNDING

| Funder | Grant(s) | Author(s) |
| --- | --- | --- |
| Shaanxi Provincial Science and Technology Department | 2022ZDYF126 | Mingxue Zheng |
| Shanxi Agricultural University | J202111305 | Xiaoling Lv |

## AUTHOR CONTRIBUTIONS

Xiaoling Lv, Conceptualization, Data curation, Formal analysis, Investigation, Writing – original draft | Liwen Wang, Data curation, Formal analysis, Software | Sichen Dong, Formal analysis, Methodology, Software | Yutong Yang, Formal analysis, Methodology | Xueqi Zhang, Data curation, Software | Tong Xu, Data curation, Software | Kailing Cui, Validation | Zhibin Niu, Validation | Wenrui Zhao, Supervision | Rui Bai, Supervision | Li Zhang, Supervision | Mingxue Zheng, Conceptualization, Resources, Writing – review and editing

## ETHICS APPROVAL

All experiments involving animals were carried out in accordance with Institutional Animal Care and Use Committee guidelines and were approved by the Animal Protection and Utilization Committee of Shanxi Agricultural University, China.

## ADDITIONAL FILES

The following material is available online.

### Supplemental Material

**Supplemental material (Spectrum00416-25-S0001.doc).** Fig. S1 and S2.

### Open Peer Review

**PEER REVIEW HISTORY (review-history.pdf).** An accounting of the reviewer comments and feedback.

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
