## [Reviewer comments · Microbiology Spectrum]

Microbiology Spectrum

***Eimeria tenella* AMA1 regulates host cell apoptosis through mitochondrial pathway and death receptor pathway**

Xiaoling Lv, Liwen Wang, Sichen Dong, Yutong Yang, Xueqi Zhang, Tong Xu, Kailing Cui, Zhibin Niu, Wenrui Zhao, Rui Bai, Li Zhang, and Mingxue Zheng

Corresponding Author(s): Mingxue Zheng, Shanxi Agricultural University

Review Timeline:

Submission Date:	February 11, 2025
Editorial Decision:	April 17, 2025
Revision Received:	May 5, 2025
Accepted:	May 19, 2025

Editor: Neil Mabbott

Reviewer(s): The reviewers have opted to remain anonymous.

Transaction Report:

DOI: <https://doi.org/10.1128/spectrum.00416-25>

Re: Spectrum00416-25 (*Eimeria tenella* AMA1 regulates host cell apoptosis through mitochondrial pathway and death receptor pathway)

Dear Prof. Mingxue Zheng:

Thank you for the privilege of reviewing your work. Below you will find my comments, instructions from the Spectrum editorial office, and the reviewer comments.

While each of the reviewers expressed interest in your study and acknowledged the importance, they have identified some areas for revision. Attention to these will significantly enhance the quality of your manuscript.

Revision Guidelines

Sincerely,
Neil Mabbott
Editor
Microbiology Spectrum

Reviewer #1 (Comments for the Author):

Avian coccidiosis, caused by protozoa of the genus *Eimeria*, represents one of the most prevalent and economically significant parasitic diseases in global poultry production. Among the causative species, *Eimeria tenella* demonstrates particularly high pathogenicity, primarily targeting the cecal epithelium. In this study, the author successfully established an in vitro model of *E.*

tenella infection using chicken embryo cecal epithelial cells. While apical membrane antigen 1 (AMA1) is conventionally known to facilitate host cell invasion through moving junction formation, this investigation reveals its novel role in triggering parasite apoptosis. The study systematically evaluates the regulatory mechanisms through which EtAMA1 modulates host cell apoptosis. The research presents compelling findings with clear logical progression and adequate language expression, though several minor revisions could further enhance the manuscript.

- In line 82, please specify the full name of ROP38. The full name of EGFR should also be clearly written in the follow text.
- In line 159, please modify "3-6 days" to "3~6 days".
- In line 185, 188, 202, please change "hour" to "h".
- In line 272, please specify the concentration of IPTG.
- In line 293, writing "at 4 h" as "at 4 h post infection" is more appropriate.
- the role of AMA1 in moving-junction should be addressed in the introduction section.
- a house keeping control should be given for the western blots.
- a negative control should be given for the IFA analyze.

Reviewer #2 (Comments for the Author):

However, they could not demonstrate that AMA1 was the key factor when they only used recombinant AMA1 protein here (is AMA1 secreted into host cell? interact with which receptor?).

The expression change revealed by qPCR of 3 genes could not reveal the profile of the two pathways. otherwise, qPCR should be used for validation after a transcriptomic analysis.

For publication in Microbiology Spectrum, those results (figures 1 and 2) for recombinant protein preparation and so on should be in supplementary data.

The writhing of this MS needs thorough revision (see the minor issue for example)

In Figure 5, key information is missing from the figure legend. Although the statistical methods are described in the Materials and Methods section, the legend should still specify the number of replicates, type of error bars (e.g., SD or SEM), the statistical test used, and the p value thresholds. Additionally, the y-axis labels in panels D-F should clearly indicate the units of measurement.

Minor issues:

Line 75 incorrectly refers to *E. tenella* as belonging to Trichoptera. A little strange why such a term occurs here. It should be classified under Apicomplexa.

Line76: "spp." should not be italicized, in accordance with taxonomic conventions.

Lines 152-164: the manuscript lacks detailed information regarding the induction conditions for EtAMA1 protein expression in both *Pichia pastoris* and *E. coli* systems. To ensure reproducibility, it is recommended that the authors specify the inducer concentration, induction temperature, duration, and the optical density (OD600) at which induction was initiated.

In Figure 3A, should add molecular weight markers to the Western blot to clearly indicate the expected size of the recombinant EtAMA1 protein.

In Figure 9, the schematic diagram lacks visual clarity and should be replaced with a higher-resolution image. The current representation may also lead to misunderstanding, as it could be interpreted that EtAMA1 promotes the expression of Caspase-8 and Caspase-3 mRNA by downregulating Fas and TNFR1.

Reviewer 1

Minor points

1. In line 82, please specify the full name of ROP38. The full name of EGFR should also be clearly written in the follow text.

Answer for the question:

We carefully proofread the article and modified it. **lines 82, 86 in the revised manuscript.**

2. In line 159, please modify "3-6 days" to "3~6 days".

Answer for the question:

We carefully proofread the article and modified it. **lines 164 in the revised manuscript.**

3. In line 185, 188, 202, please change "hour" to "h".

Answer for the question:

We carefully proofread the article and modified it. **lines 202, 204, 212, 218, 219 in the revised manuscript.**

4. In line 272, please specify the concentration of IPTG.

Answer for the question:

We added concentration of IPTG in the revised manuscript. **lines 177 in the revised manuscript.**

5. In line 293, writing "at 4 h" as "at 4 h post infection" is more appropriate.

Answer for the question:

We carefully proofread the article and modified it. **lines 307 in the revised manuscript.**

6. the role of AMA1 in moving-junction should be addressed in the introduction section.

Answer for the question:

We added the role of AMA1 in moving-junction into the introduction section. **lines 87-93 in the revised manuscript.**

7. a house keeping control should be given for the western blots.

Answer for the question:

EtAMA1 is an exogenous gene in chickens. We express it using a yeast expression system. Therefore, when the Western blot results show a specific band at approximately 58 kDa, matching the predicted size, it indicates successful expression of *EtAMA1* in yeast. Based on this, house keeping control was not used in the western blots experiment to identify *EtAMA1* protein expression.

8. a negative control should be given for the IFA analyze.

Answer for the question:

We have added a negative control for IFA analysis in the revised manuscript based on the suggestion.

Reviewer 2

Major issues:

1. However, they could not demonstrate that AMA1 was the key factor when they only used recombinant AMA1 protein here (is AMA1 secreted into host cell? interact with which receptor?).

Answer for the question:

In this study, it was indeed unable to confirm that *EtAMA1* is a key molecule involved in the regulation of host cell apoptosis by *E. tenella*. However, the experimental results confirm that *EtAMA1* can regulate host cell apoptosis through mitochondrial pathway and death receptor pathway. Therefore, we did not include a description in the manuscript that *EtAMA1* is a key molecule regulating host cell apoptosis.

2. The expression change revealed by qPCR of 3 genes could not reveal the profile of the two pathways. otherwise, qPCR should be used for validation after a transcriptomic analysis.

Answer for the question:

We strongly agree with your statement that the expression changes of the three genes revealed by qPCR cannot reveal the characteristics of these two pathways. Therefore, we simultaneously used the WB method to detect protein expression levels (**original Figure 8E, now shown in revised Figure 6E**) . We are very sorry that our wording has caused you such a misunderstanding. We have revised the description of the results in this section of the revised Figure legends.

3. For publication in Microbiology Spectrum, those results (figures 1 and 2) for recombinant protein preparation and so on should be in supplementary data.

Answer for the question:

We have placed Figure 1 and Figure 2 in the supplementary Figures according to the suggestion.

4. The writhing of this MS needs thorough revision (see the minor issue for example)

Answer for the question:

We carefully proofread the article and modified the errors. **lines 75, 76, 165-173, 176-181 in the revised manuscript.**

5. In Figure 5, key information is missing from the figure legend. Although the statistical methods are described in the Materials and Methods section, the legend should still specify the number of replicates, type of error bars (e.g., SD or SEM), the statistical test used, and the p value thresholds. Additionally, the y-axis labels in panels D-F should clearly indicate the units of measurement.

Answer for the question:

We have added explanations in the legend regarding the number of repetitions, types of error bars, statistical tests used, and p-value thresholds according to the suggestion. Figure 5D-F (Figure 3D-F in the revised Figures) show the detection of Caspase-8 and Caspase-9 activity using ELISA method. There is no unit to express the level of activity based on the OD value

Minor issues:

1. Line 75 incorrectly refers to *E. tenella* as belonging to Trichoptera. A little strange why such a term occurs here. It should be classified under Apicomplexa.

Answer for the question:

We carefully proofread the article and modified the errors. **lines 75 in the revised manuscript.**

2. Line76: "spp." should not be italicized, in accordance with taxonomic conventions.

Answer for the question:

We carefully proofread the article and modified the errors. **lines 76 in the revised manuscript.**

3. Lines 152-164: the manuscript lacks detailed information regarding the induction conditions for EtAMA1 protein expression in both *Pichia pastoris* and *E. coli* systems. To ensure reproducibility, it is recommended that the authors specify the inducer concentration, induction temperature, duration, and the optical density (OD₆₀₀) at which induction was initiated.

Answer for the question:

We have added detailed information on the induction conditions of *EtAMA1* protein expression in *Pichia pastoris* and *E. coli* systems in the new manuscript. **lines 165-173, 176-181 in the revised manuscript.**

4. In Figure 3A, should add molecular weight markers to the Western blot to clearly indicate the expected size of the recombinant *EtAMA1* protein.

Answer for the question:

Thank you for your suggestion. We have added molecular weight markers in the Western blot to clearly indicate the expected size of the recombinant *EtAMA1* protein (**original Figure 3A, now shown in revised Figure 1A**).

5. In Figure 9, the schematic diagram lacks visual clarity and should be replaced with a higher-resolution image. The current representation may also lead to misunderstanding, as it could be interpreted that *EtAMA1* promotes the expression of Caspase-8 and Caspase-3 mRNA by downregulating Fas and TNFR1.

Answer for the question:

The statement in Figure 9 may indeed lead to misunderstandings. Therefore, we have decided to delete the schematic diagram after careful consideration.

Re: Spectrum00416-25R1 (*Eimeria tenella* AMA1 regulates host cell apoptosis through mitochondrial pathway and death receptor pathway)

Dear Prof. Mingxue Zheng:

Thank you for resubmitting your revised manuscript. I am content with these revisions and it is my opinion that the issues raised by the Reviewers have been satisfactorily addressed.

Your manuscript has therefore been accepted for publication, and I am forwarding it to the ASM production staff for publication. Your paper will first be checked to make sure all elements meet the technical requirements. ASM staff will contact you if anything needs to be revised before copyediting and production can begin. Otherwise, you will be notified when your proofs are ready to be viewed.

Sincerely,
Neil Mabbott
Editor
Microbiology Spectrum